# A Computer Vision System for Staff Gauge in River Flood Monitoring

**Luisiana Sabbatini** [1,*] , **Lorenzo Palma** [1] , **Alberto Belli** [1] , **Francesca Sini** [2] **and Paola Pierleoni** [1]

1   Department of Information Engineering, Università Politecnica delle Marche, Via Brecce Bianche 12, 60121 Ancona, Italy; l.palma@univpm.it (L.P.); a.belli@staff.univpm.it (A.B.); p.pierleoni@univpm.it (P.P.)
2   Regione Marche—Servizio Protezione Civile, Centro Funzionale Regionale, Via di Colle Ameno 5, 60126 Ancona, Italy; francesca.sini@regione.marche.it
*   Correspondence: l.sabbatini@pm.univpm.it; Tel.: +39-071-2204128

**Abstract:** Rivers close to populated or strategically important areas can cause damages and safety risks to people in the event of a flood. Traditional river flood monitoring systems like radar and ultrasonic sensors may not be completely reliable and require frequent on-site human interventions for calibration. This time-consuming and resource-intensive activity has attracted the attention of many researchers looking for highly reliable camera-based solutions. In this article we propose an automatic Computer Vision solution for river's water-level monitoring, based on the processing of staff gauge images acquired by a V-IoT device. The solution is based on two modules. The first is implemented on the edge in order to avoid power consumption due to the transmission of poor quality frames, and another is implemented on the Cloud server, where the frames acquired and sent by the V-IoT device are processed for water level extraction. The proposed system was tested on sample images relating to more than a year of acquisitions at a river site. The first module of the proposed solution achieved excellent performances in discerning bad quality frames from good quality ones. The second module achieved very good results too, especially for what it concerns night frames.

**Keywords:** computer vision; river flood; Visual-IoT

## 1. Introduction

Despite technological advancements and efforts, river basins are often source of damages and risks, especially for the population living in the neighboring areas. The risk related to flash flood, which can be valued based on multiple factors such as local hazards, exposure, vulnerability, and emergency and recovery capabilities [1], is increasing over the time and it is envisioned to get even worse due to climate change and atmospheric warming [2]. Several researchers attempted to develop models able to predict future flood hazards, highlighting also the correlation with climate changes [3,4], but it is hard to develop area specific adaptation of such models. For this reason most of the time local public entities are predisposed to control the territory in which continuous monitoring mechanisms for the safeguard of citizens are used. In particular, continuous monitoring systems are generally set up making mandatory the human supervision and contribution, thus consuming strengths and resources that public administration has at its disposal, from both an economic and workforce point of view. Despite the costs associated to such kind of monitoring mechanisms, the societal benefits have been estimated as higher [5]. However, to cope with such a source of inefficiency and waste identified, it is necessary to envision and implement smart and automatic solutions, by exploiting innovative technologies as Information Technologies, Sensing Technologies, and Artificial Intelligence, as some researchers already tried to address.

Automatic monitoring solutions are at the base of developing smart Early Warning (EW) systems for flood hazards, which usually exploit, among relevant input data, the

water level of rivers [6,7]. Nonetheless, the solutions proposed in the literature and focused on EW do not go into the details of how the water level is measured, apart from saying the type of instrumentation adopted [8,9]. Commonly adopted technologies for monitoring river flood are pressure transducers, rangefinders, ultrasonic, radar as well as optical sensors [10]. Some of these technologies require frequent calibration, otherwise, the accuracy becomes very low when objects like wooden logs pass underneath, or when the wind causes waves. Moreover, these technologies are prone to measurement errors which could happen especially during dry riverbed and during extreme weather conditions like heavy rainfall, which are those conditions to be controlled more strictly for flood monitoring purposes. This has led many public entities to set up low-cost cameras for remote monitoring and visual inspection of river sites. In such sites the cameras can also frame the staff gauge which is generally installed to indicate the water level of the river. Using such cameras able to provide images related to the water level, it is possible to develop a Computer Vision (CV) system for flood hazards. Therefore, many researchers in recent years focused on using optical sensors like cameras to monitor the water level [10]. A camera equipped with remote transmission capability and eventually processing capability can be used as Visual-IoT device (V-IoT) for estimating the water level. Developing such V-IoT systems require less effort for calibration respect to using sonic and radar sensors, is economically viable, and allows the creation of a reliable sensing network, capable of being exploited for automatic monitoring solutions and on-demand remote visual inspection of river sites. For these reasons the application of V-IoT devices is spreading over the time, but related criticalities should be taken into account when evaluating the creation of such a monitoring system [11]. In more details, when dealing with V-IoT systems, data transmission, processing and storage have a huge impact in terms of available bandwidth and energy consumption. V-IoT systems produce huge amounts of data in relatively short time, forcing to be mindful in managing them. In order to mitigate the transmission and storage constraints, Peng et al. [12] proposed an enhancement method to overcome the false contour and color distortion connected with bit-depth compression, given that this is a widely used solution. As a general suggestion, Ji et al. [13] highlighted the importance of optimising transmission, storage and processing, based on the specific system tasks to be completed, taking into consideration delay sensitive and context aware video services. Moreover, when V-IoT nodes are installed in natural contexts, like those of relevance for flood monitoring purposes, they are generally battery-powered thus forcing to be thrifty in the use of energy required for data transmission.

In the flood monitoring scenario the challenge is to develop a completely automatic solution capable of working 24 h a day in unconstrained natural landscape with possibly adverse weather conditions, and of easily adapting to multiple sites with easy to set adjustments. Focusing on the image processing side, Yang et al. [14] presented the development of a system tested in an indoor controlled laboratory, simulating rain. The development and testing configuration makes the proposed solution hardly applicable in unconstrained natural landscape, at least to our real scenario. Kim et al. [15] developed a method and tested it on 4 real sites. The proposed solution based on CCTVs is not able to work during night and two different approaches for computing the water level are proposed, one based on frame differencing, and another based on Optical Character Recognition (OCR). By looking at some sample frames, we noticed that during extreme weather the quality of captured images might become very low, thus making OCR not a viable nor reliable solution. Noto et al. [16] proposed a solution tested in a non-simulated scenario with some critical conditions of lighting and landscape. Their solution is capable of working 24 h a day but adopted a pole instead of a gauge. Moreover, water level estimation relies on the assumption that the pole is brighter than anything in the Region of Interest (ROI) framed, making this solution not versatile at all. Similarly, in [17] it is presented an automatic solution for detecting gauge's ROI, but it exploits the color and morphology of a specific kind of gauge, which is not the one commonly installed on generic sites. Zhang et al. [18] described a system whose performances have been tested during complex conditions.

Main limitations of the proposed solution are connected to the gauge's ROI identification, which is not automatic and implies ROI setting during installation, but doesn't ensure reliability on the long run in case of camera motion. In fact, working in unconstrained natural landscapes, it is also necessary to use a solution capable of detecting the gauge's ROI if it changes due to any misalignment of the camera. In [19] the Authors proposed a 24 h a day monitoring system where the ROI is once again manually set. The solution is then tested on two sites, focusing on processing steps targeted at the extraction of the water level even under complex illumination conditions. Similarly, Hies et al. [20] proposed a solution capable of working h24 thanks to infrared camera, but the identification method of the gauge's ROI is manual. Royem et al. [21] proposed a method for one site, using one static ground camera. The proposed method is strongly dependent upon calibration procedures. Moreover, the gauge's ROI selection is done through color-based processing and requires a gauge that is chromatically distinguishable from the context in which it is immersed. Hasan et al. [22] presented a h24 monitoring solution, tested in multiple sites. In order to select the ROI of the gauge they installed a big white board near to it. In [23] the proposed solution uses multiple reference points to select the ROI, which is very ample respect to the contained gauge. Moreover, the system has not been tested during night. In our scenario, the gauges are already installed but it is advisable to avoid installing similar reference points. Bruinink et al. [24] proposed a portable solution, that can be implemented in mobile phones The solution has been successfully tested in 9 sites, but the gauge's ROI detection is performed using a textons-based approach which is not versatile at all. In fact, this approach is usable only when the gauge is highly distinguishable from the background, which is not always a true hypothesis. Jafari et al. [25] propose an advanced method which exploits CNN and leverages time-lapse photos and object-based image analysis. The methodology reaches very good performances in both the laboratory and two field experiments, nonetheless it relies on a strong site-specific adaptation phase. Our aim is to propose a more versatile and adaptable solution. Isidoro et al. [26] propose a water surface measurement through image processing, still the proposed flowchart is quite easy and do not guarantee successful scalability to several diversified sites.

Concentrating on cases where the gauge is framed by a static ground camera, in this paper we present a CV system for monitoring river flood based on the processing of images of the gauge installed in the rivers. In particular, our aim is the creation of a versatile and smart automatic water level measurement algorithm, to be applied in multiple sites with minimal adaptation effort, thus ensuring its scalability in the near future. By looking at past works, there are only attempts of automatizing the water level computation task, some of them reaching results in laboratories, or failing under complex conditions, which are very frequent in real river sites and should be addressed. Based on the reviewed literature we can define the common steps of such a solution, but we cannot find one example that perfectly suits our scenario. Given the required versatility, we believe that advanced solutions based on Machine Learning (ML) are better than traditional Image Processing techniques. However, we must point out that ML based solutions can be energy intensive, just like the transmission of images or videos. Therefore, we will present an important basic, fast and computationally light image processing methodology, capable of testing the quality of the image collected by the installed camera. This procedure becomes essential for V-IoT systems installed in unconstrained environments, even more so in those cases where energy consumption due to images transmission is a constraint. Our intention is to develop a versatile automatic solution, able to work in different sites, after a light and easy adaptation at the moment of installation. The presented works have pros and cons with respect to our requirements, and for this reason we developed a new approach, mixing the strengths of the reviewed literature in order to develop a smart, flexible, scalable and reliable solution. The reviewed works are usually very site-specific, or incomplete respect to the identified requirements.

Specifically, the value of the paper is manifold. To begin with, recent literature about CV-based water level monitoring algorithms is reviewed. In addition, a computationally

light (hence, suitable for edge implementation) algorithm, for ensuring the quality of collected data to improve performance and durability of V-IoT systems immersed in fully unconstrained natural environment, is proposed. Lastly, the entire solution proposed is described in details and its relation with existing solutions and its virtues are highlighted.

In the next Section we are going to describe the scenario in which we are working, together with the experimental setup used for developing the prototype solution. After testing the proposed system, we will highlight criticalities to be taken into consideration for improving it and making it suitable for our objective.

## 2. Materials and Methods

In this paper we propose an automatic CV solution capable of detecting and computing the water level of a river, taking as input a frame snapped by a V-IoT device. The proposed solution has been created with the following requisites in mind, set together with the entity in charge of disaster management:

a. be fully automatic;
b. be able to detect the water level with an accuracy of $\pm 3$ cm;
c. require minimum site-specific customization, except for the initial in-site installation;
d. be able to work day and night;
e. be reliable even during extreme weather conditions;
f. transmit to the central server high-quality data only;
g. be able to work in sites where the gauge is made up of multiple pieces, framed all together by the camera.

To develop and test the proposed CV solution, we use a data set composed of more than 10 thousands sample images collected using a cam framing the river gauge in one specific site of interest. The camera and the gauge were immersed in unconstrained environment, which increases the complexity of the proposed solution. The camera is installed in Senigallia (AN), Italy, near Garibaldi's Bridge, depicted in Figure 1 which has been captured during a day by the camera used for developing our solution, but using a different set-point for the acquisition respect to the one adopted for capturing frames of the gauge. From the picture it can be seen that the camera is installed inside a urban area, but during the night it does not require artificial illumination to work, thus being an hardware suitable for installation in completely natural sites where there is no artificial light available and only the IR could be exploited.

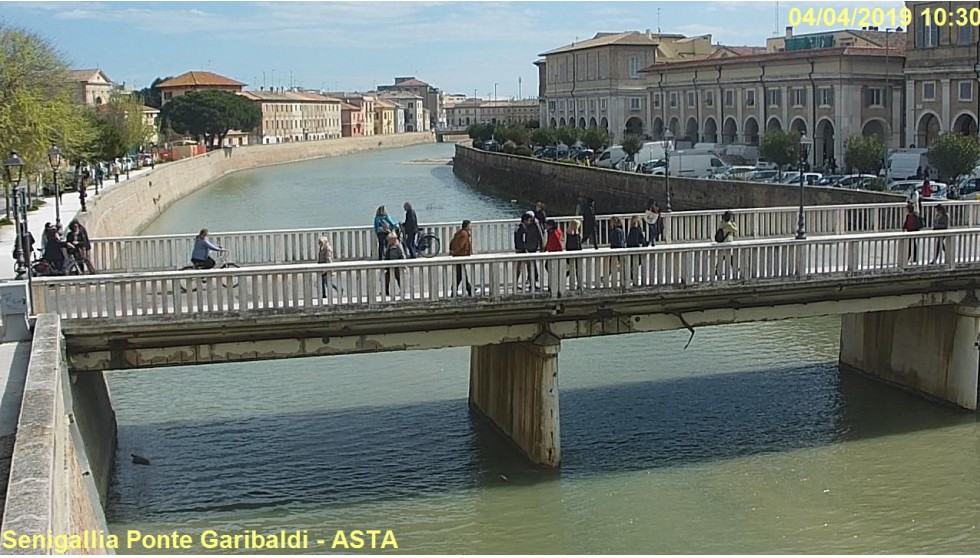

**Figure 1.** Panoramic view on the acquisition site.

The data collection architecture used indeed as static ground camera a Dahua DH-SD50C120S 1.3 Megapixel RGB 1280 × 960, with Auto ICR for day/night, and TCP/IP

connection. The cam snapped one frame every 30 min and these data were uploaded on a dedicated server, which made them remotely accessible. All the acquired frames are RGB, even though night frames seems gray-scale due to the fact that they are collected through the auto ICR function. Data were collected for more than a year, resulting in a collection of frames with almost every shade of picture linked to different seasons, light and weather conditions. Specifically, we have six extreme weather conditions events, resulting in more than twenty related frames. Going deeper into the data collected, night frames are quite similar to each other, while daily ones present differences caused by the different inclination of sun's rays. In Figure 2 we show the two main typologies of frames collected by the cam, day and night ones. Moreover, during the year, some samples were overexposed frames, and some others were blurred or bad quality ones due to extreme weather conditions, like the examples shown in Figure 3.

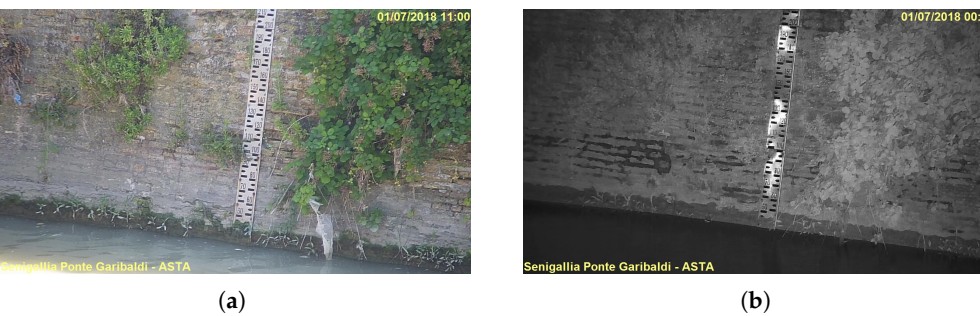

|  (**a**)  |  (**b**)  |

**Figure 2.** Example of two typical images collected by our system. (**a**) Day frame, collected by the cam 01/07/2018 at 11:00 a.m.; (**b**) Night frame, collected by the cam 01/07/2018 at midnight.

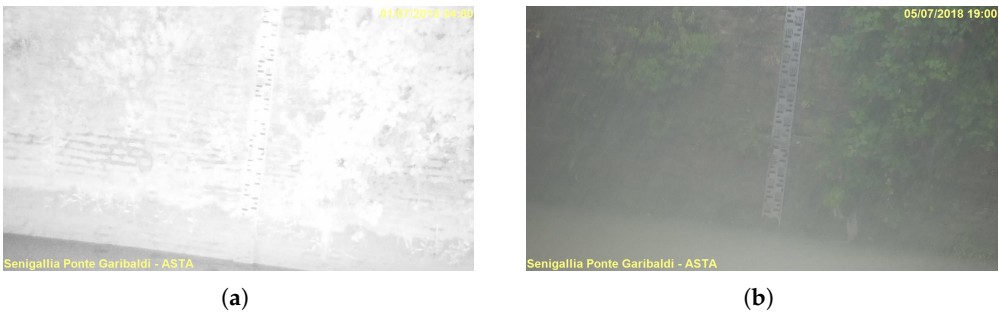

|  (**a**)  |  (**b**)  |

**Figure 3.** Example of overexposed and bad quality frames snapped by the camera. (**a**) One of the overexposed frames; (**b**) One example of excessively blurred frame.

The proposed solution must be able to process different typologies of frames collected by the cams, and for this reason we are going to present a smart solution able to understand which kind of frame is the one under analysis, and to perform tailored computations to extract the water level. In order to compute the water level during the entire day with an accuracy of ±3 cm, image processing should rely on good quality images but it is easy to have some bad quality frames, either due to bad weather or to contingencies connected to the unconstrained and non-standardised context of use. Therefore, we worked on a specific module of the solution, to be implemented on the edge, which processes each collected frame as soon as it is snapped with basic image processing algorithms, with the aim of ensuring the collected frame is a good quality one. This module takes as input a frame snapped by a V-IoT device, and classifies it as either day, night, or bad quality (either overexposed, blurred, or with weather related artifacts), through light and fast computations suitable for edge implementation. In the proposed solution the good quality night frames, are pre-processed at the edge and only the grayscale version is sent to the server, while day frames, being more complex and less standardised, are sent in their original form. Specifically, for what it concerns night frames, only the image compressed in one color channel is sent to the cloud where the Water Level Computation

module is implemented. By adding Image Quality Check at the edge, we are going to positively impact both data transmission and energy consumption of each of our V-IoT nodes, improving the overall reliability and performance of the network and system.

In the following sub-sections we are going to describe in detail each of the solution modules, and to specify whether they are suitable for edge or cloud computing.

### 2.1. MODULE 1: Image Category Classification

We started cleaning available data, 16,124 frames, from those samples not usable due to a different camera set point, which resulted in frames capturing the nearby landscape instead of the gauge. Exploring the resulting data set after upfront cleaning, we kept 13,422 samples from the original set, and we looked for quantitative, metrics able to differentiate the distinct types of frames. We evaluated the following metrics, that are going to be called from now on with the synthetic abbreviation in brackets:

- Mean of all pixels of RGB color channels (MNall);
- Mean of the saturation channel of the image converted into HSV color model (MNs);
- Root Mean Square of RGB channels (RMSall);
- Root Mean Square of saturation channel (RMSs);
- Maximum inter-pixel difference, computed as the maximum along all the pixel intensities, minus the minimum (delta);
- Variance of the image histogram (VARih);

We then evaluated the correlation between each of these metrics, and the image classes, finding some interesting strong relations. Specifically, MNall is strictly discerning the overexposed frames from the other classes. MNs and RMSs were strongly different between day frames and the other classes, as could be expected since during the night the camera acquires through auto-IR cut filter and the resulting RGB image is like a gray-scale one. The night frames have very low saturation thus resulting very similar to frames snapped during bad weather or in case of bad quality images, where the saturation gets low too. Another interesting connection has been found for the delta metric, suitable for discerning between bad quality, against good quality both day or night frames.

In order to develop a broad reference for understanding differences in the metrics between the distinct frame classes we plotted the three most discerning variables, specifically MNall, RMSs, and delta, by group, obtaining the plot shown in Figure 4.

By analysing the 3D plot, we found that group of images belonging to the same class, are quite well separable by using the three metrics plotted. Based on the analysis of the entire clean data set, we can obtain a quantitative measure of frame class, distinguishing between night, day, and bad quality. Therefore, we developed a multi-threshold based algorithm for discerning the class of the frame. The algorithm is integrated into a specific module able to discern bad quality frames, from good quality day and night ones, thus allowing us to send only images from which it is possible to obtain useful information regarding the water level. This module should be implemented on the edge and is especially useful for battery-powered V-IoT devices, given that it was designed to save the energy consumption due to the transmission of unusable data. The proposed module is illustrated in the flowchart shown in Figure 5, where the choice of the multi-threshold based algorithm for discerning frame category is due to its fast response and low computation load.

Once the module for image category classification analyses one snapped frame and understands its category, the frame may be classified as night, day, or bad quality. Bad quality frames, which can be connected to different kinds of causes, are discarded, whereas day and night frames are sent to the central server for further processing. In particular, in case of a day frame, the module sends directly the image to the central server. If the frame is a night one, the module sends directly the gray-scale image to the server, thus reducing the size of transmitted and stored data: from three color channels, to one.

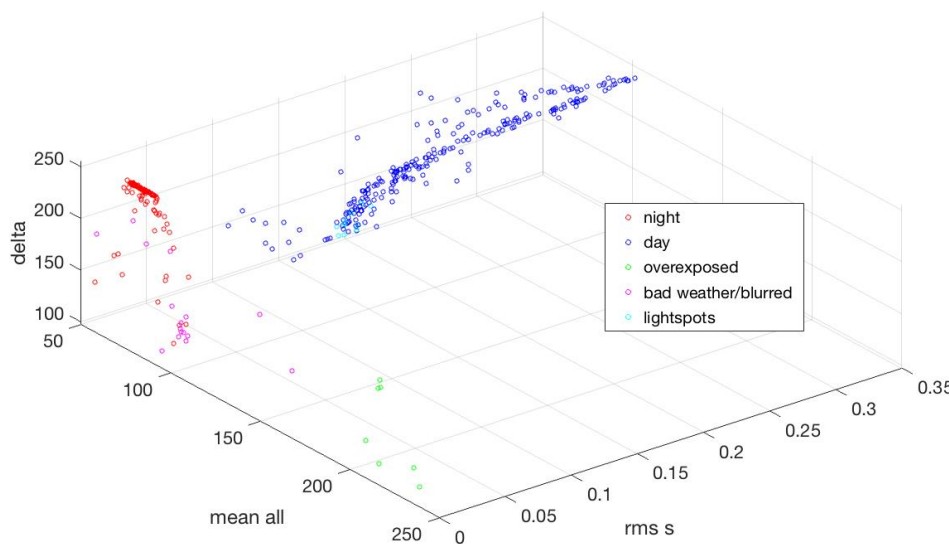

**Figure 4.** 3D plot of MNall, RMSs, and delta, coloured by group as specified in the legend.

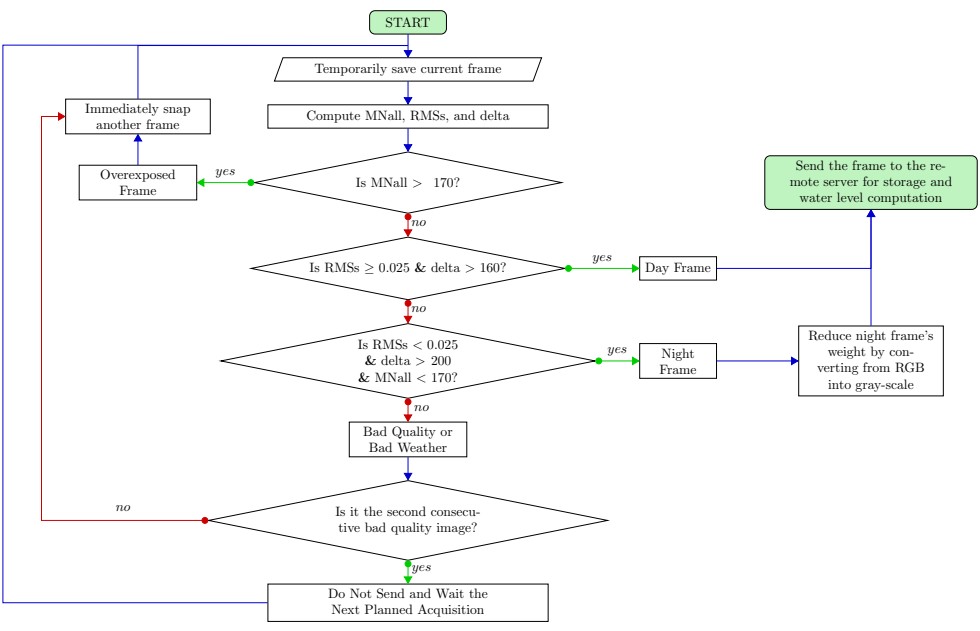

**Figure 5.** Flowchart of the image category classification algorithm.

## 2.2. MODULE 2: Gauge Detection and Water Level Computation

Once an image is sent by the V-IoT device to the server, it is temporarily saved for being processed by the second module. It is composed of two parallel algorithms, one for daily frames, and another for night ones. The first step is common to both day and night frames, and concerns the ortho-rectification of the image. Through this step, we pass from an image where the gauge is distorted, due to the relative perspective of the camera and the gauge, to a rectified image where the gauge seems frontally framed. Specifically, this procedure has to be set for each site, and starts with the identification of the four rectangle's vertices and the rectified rectangle associated to a perfect frontal view, as shown in Figure 6. Once associated each corresponding vertex with its "correct" frontal position, image rectification can be performed. This solution is more than a simple rotation, and requires some portion of the image to be filled with a value since those pixels are not

present in the original frame. As can be noticed by looking at Figure 7, in the rectified image, the water level is an horizontal line, thus making the analysis easier.

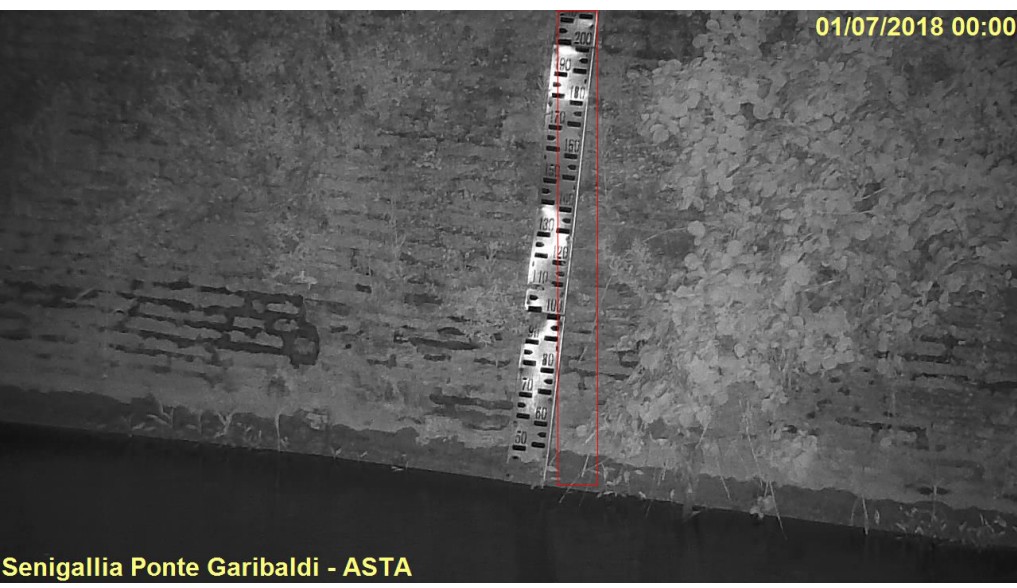

**Figure 6.** The original frame with the desired gauge positioning, used to compute image rectification settings.

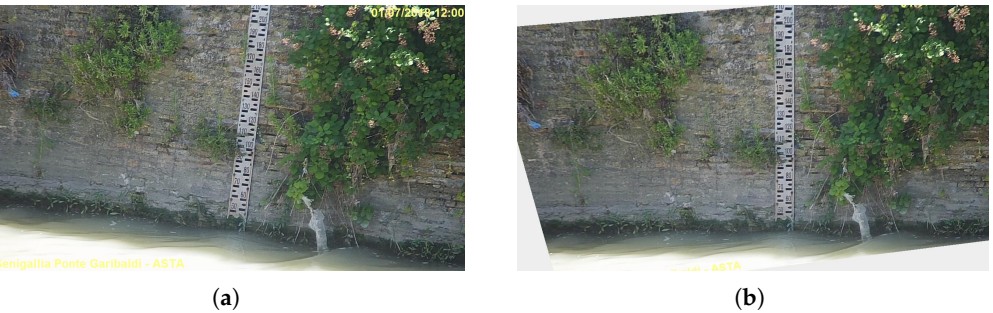

(**a**)                                                                 (**b**)

**Figure 7.** Image ortho-rectification example. (**a**) The original frame sent by the V-IoT device; (**b**) The frame after ortho-rectification based on specific site's setting.

In order to set the ortho-rectification parameters we used one sample frame. Once defined, these parameters are fixed for the site under analysis and should be updated only if the system setup changes.

Then, if the image under analysis was classified as a daily one, the processing steps after rectification are the following:

d1.    top hat filtering, using as structuring element a disk of 15 pixels radius;

d2.    adjust image intensities, saturating top and bottom 1% of all pixel values;

d3.    binarize the image using a fixed threshold (45);

d4.    eliminate from the binary image those connected regions having area lower than 50 pixels, to reduce noise;

d5.    fill the holes;

d6.    perform morphological closing using as structuring element a line of 30 pixels;

d7.    compute the percentage of white pixels for each row over the columns;

d8.    compute the adaptive threshold as mean minus one standard deviation of the row percentages;

d9.    start from the bottom and find the first line where the row percentage exceeds the threshold, which is the water level;

d10.  draw a red line corresponding to the computed water level.

In Figure 8 we show the outcome associated to each of the listed steps, for the sample image already shown in Figure 7. Additionally, even the plot of row percentages metric, used to find the water level.

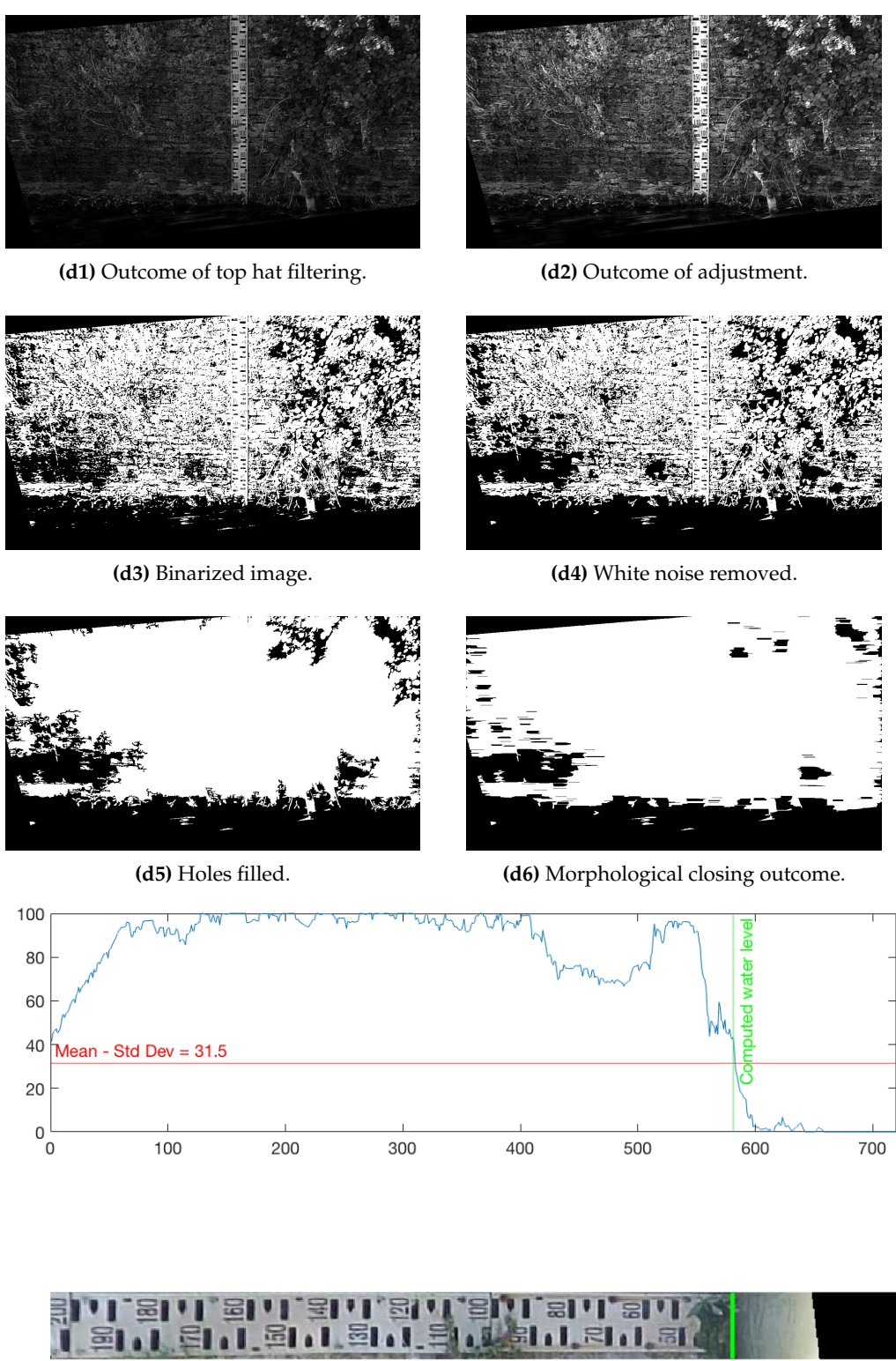

(**d1**) Outcome of top hat filtering.

(**d2**) Outcome of adjustment.

(**d3**) Binarized image.

(**d4**) White noise removed.

(**d5**) Holes filled.

(**d6**) Morphological closing outcome.

**Figure 8.** Day frames processing steps outcomes.

If the image was a night one, the processing steps after image rectification are:

n1.  detection of the gauge and cutting the image;
n2.  median filtering the retained portion;
n3.  extraction from it of 5 thresholds of intensity;
n4.  sharpening by a factor of 1.4;
n5.  clustering based on multiple thresholds computed before;
n6.  clusters' edges extraction, using Canny algorithm;
n7.  holes filling;
n8.  morphological closing using as structuring element a rectangle of 4 by 15 pixels;
n9.  eliminate from the binary image those connected regions having area lower than 50 pixels, to reduce noise;
n10. holes filling;
n11. compute the sum of black pixels for each row;
n12. assign to each row the value of 0 if the number of black pixels is lower than 70% of row pixels, 1 otherwise;
n13. find the water level which is the first non-zero line.

In Figure 9 we show the outcome associated to each of the steps, for a sample night image sent by the V-IoT device to the central server. The algorithm grounds on the fact that the night frames are very similar to each other: the water is darker than anything else, while the gauge is brighter, allowing us to compute the water line through simple computations. For what it concerns step n1, we decided to train an Aggregated Channel Features (ACF) [27] object detector on a small set of rectified images. This decision arose from the fact that due to wind and small relative movements between the rod where the cam is fixed and the gauge framed, the position of the gauge is not fixed over the time. For this reason, we are going to use the custom ACF object detector for its easy training, accuracy, and speed of use [28]. Through this detector, we are sure to reduce the ROI coherently with the position of the gauge in the image under analysis.

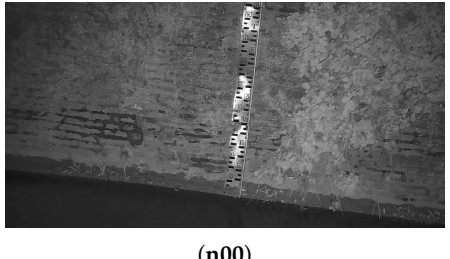
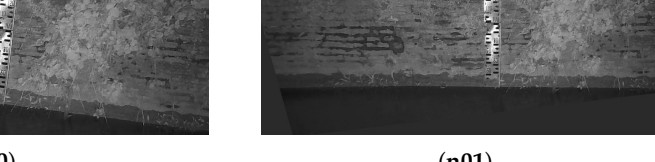

(n00)    (n01)

**Figure 9.** *Cont.*

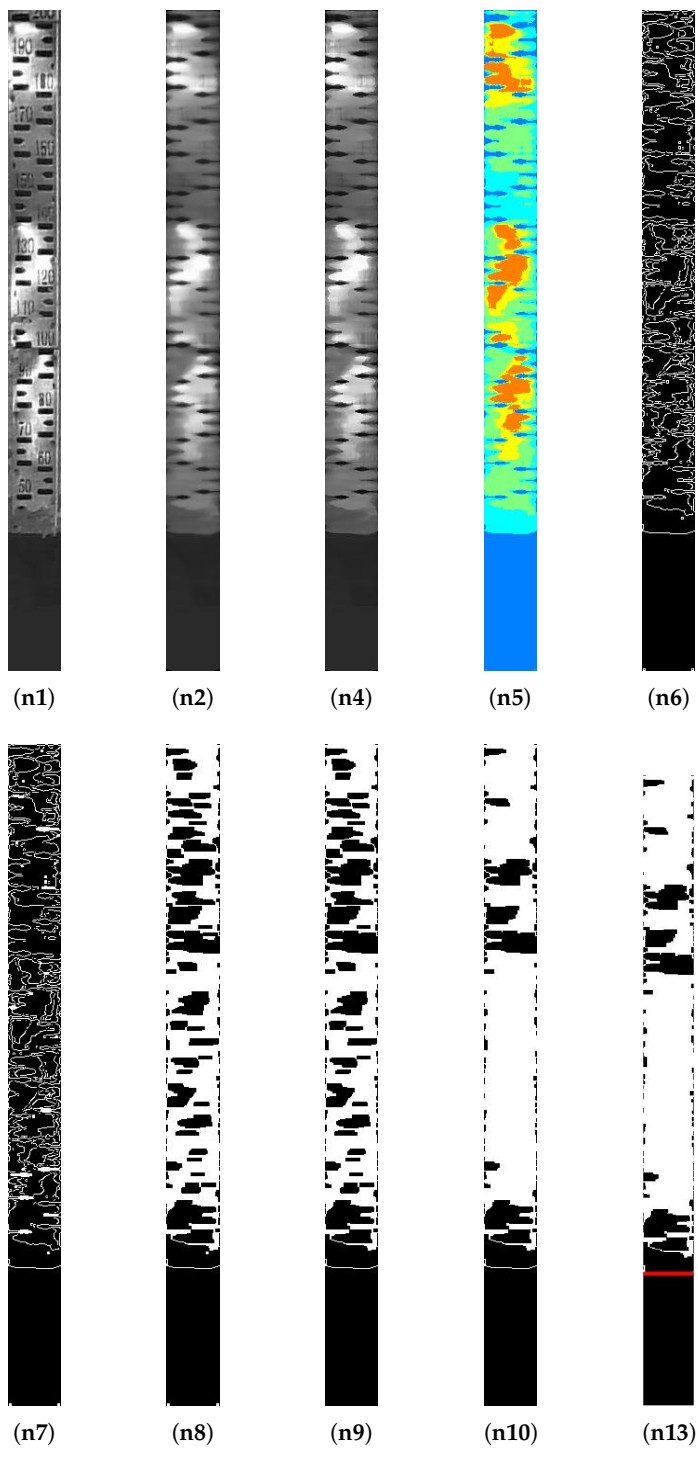

**Figure 9.** Night frame processing steps outcomes: (**n00**) Original night image in gray-scale; (**n01**) Outcome of adjustment; (**n1**) gauge's ROI detected by the ACF detector; (**n2**) after median filtering; (**n4**) after sharpening; (**n5**) Clusters based on multi-thresholds; (**n6**) Edges extracted; (**n7**) holes filled; (**n8**) morphological closing; (**n9**) small connected components removed; (**n10**) fill the holes; (**n13**) water level computed.

## 3. Results

We are going to present results achieved by each of the modules, using the data set previously described. In more details, among the 13,422 frames, we have 532 bad quality, 6248 night, and 6642 day frames. We are going to present both the results in correctly

classifying one frame as day, night or bad quality, and then the performances in computing the water level through the dedicated algorithms for respectively day and night frames. Respect to the original data set, we removed some images connected to camera set point errors (not framing the gauge, by the nearby landscape), and some others connected to the non-optimal system set up. Based on these two categories of frames, that are not usable for the sake of computing the water level, we defined improved guidelines for the installation of the V-IoT devices in other sites of interest.

To begin with our first module, among all the available frames, our algorithm found 487 of them to be low-quality. By screening these frames, we actually noticed that 464 were non-optimal, and some unusable even by the human eye, like the two shown in Figure 10. The remaining 23 bad quality images were overexposed like the one previously shown in Figure 3, hence not analysable. For what it concerns images classified as either day or night frames of good quality, 24 day and 21 night frames are not good quality at all, while all the detected as day frames are actually day, the same for night frames. Removed those that turned out to be of bad quality, the frames categorized as either day or night, were all correctly classified and hence processed each by the dedicated algorithm. Results regarding the first Module are summarized into Table 1.

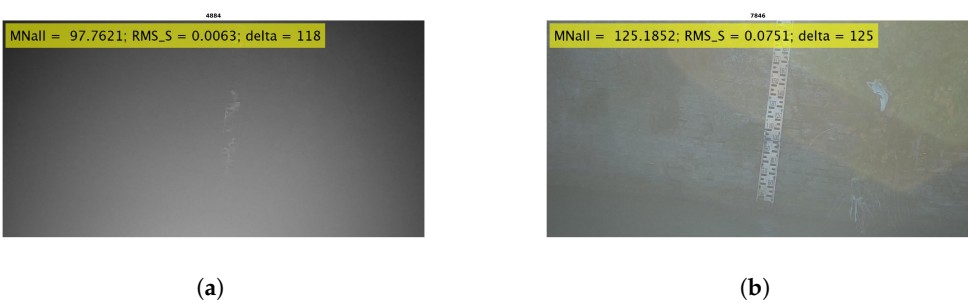

|  | (a) | (b) |

**Figure 10.** Examples of bad quality images rejected by the first Module. (**a**) One of the worse images; (**b**) Another low quality image.

**Table 1.** Summary of the Results of Module 1.

|  | Actual | Detected |
| --- | --- | --- |
| Night | 6628 | 6652 |
| Day | 6348 | 6369 |
| Bad Quality | 532 | 487 |

To present the results of the second Module, we decided to divide between "correct water level", if the computed line is no more than 3 cm apart from the actual water line, "small errors" if the computed line is 3 cm to 10 cm apart from the actual water line, and "heavy errors" otherwise. The frames with the computed water level shown have been compared with their original version and visually inspected, with the aim of assessing the goodness of computed water level. Focusing on night frames, 6545 have been further analysed to compute the water level with success, other 35 resulted in small errors, and in 48 the algorithm made heavy errors. Focusing on day frames, 5300 have been further analysed and the computed water level was correct, 465 were the small errors, and in 483 frames the algorithm made heavy errors. We would like to specify that among heavy errors, we have two samples for which the algorithm made a big mistake, as shown in Figure 11. Anyway, the fact that only two times we get this kind of error let us think that it might be caused by a wrong threshold. We therefore expect from more advanced solution, like a semantic segmentation network, to be more reliable. Results of Module 2 are summarized into Table 2.

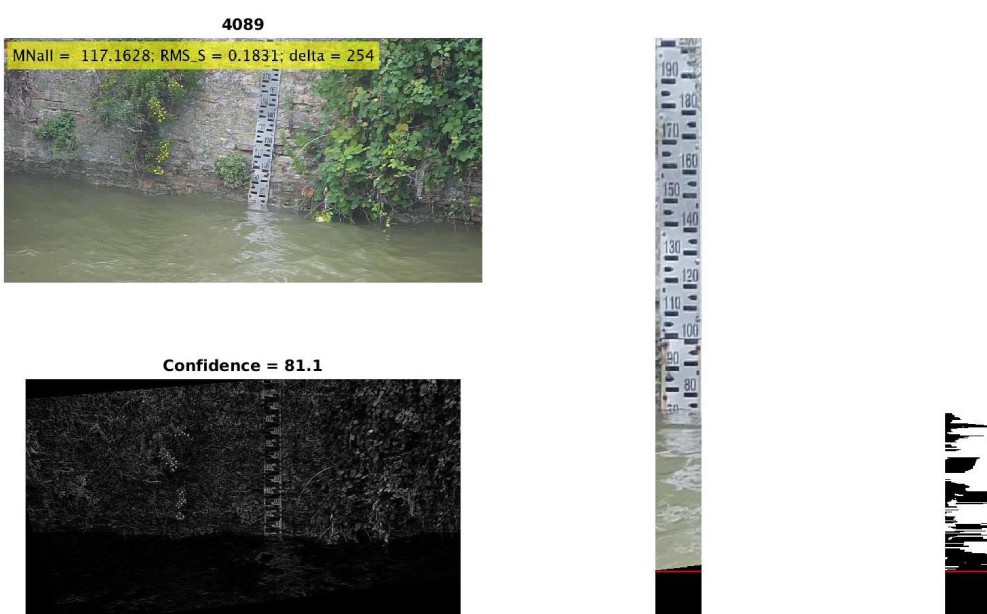

**Figure 11.** One of the two big errors done by the algorithm.

**Table 2.** Summary of the Results of Module 2.

|       | Correct Water Level | Small Errors | Heavy Errors |
|-------|---------------------|--------------|--------------|
| Night | 6545                | 35           | 48           |
| Day   | 5300                | 465          | 483          |

Going deeper into the scalability requirement for our solution, we tested the algorithm on additional frames related to other sites. Specifically, we tested the algorithm without any kind of adaptation to thirteen sites, each very different from the others. Some sites are characterised by a multi-pieces gauge, other are characterised by a very high staff gauge, thus being a small but very diversified data set. Results achieved show very good capability of being applied in those sites with similar perspective of camera and gauge, like those presented in Figure 12.

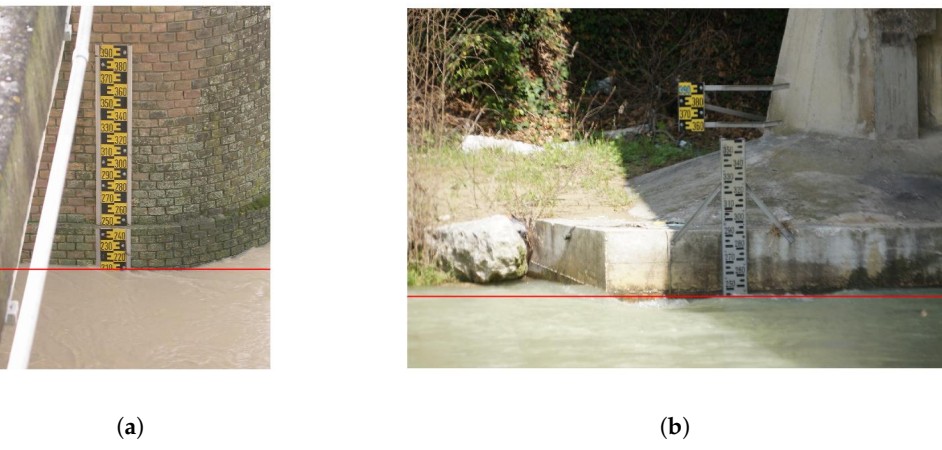

(**a**)                                              (**b**)

**Figure 12.** Results achieved for two of the additional sites analysed. (**a**) Frame of Misa River and resulting water level; (**b**) Frame of Chienti River with computed level.

## 4. Discussion

Results achieved suggest us that we can move to the next step: validation in the field, in multiple sites. During data set development phase an interval of 30 min between frames has been adopted. Given the tested reliability of Module 1, which allows to be sure about the quality of frames sent to the central server, thus optimizing transmission bandwidth utilization, we plan to put the system in place setting 3 min interval between acquisitions. We expect to be sensitive to sudden flood events through the more frequent frame snapping, without burdening transmission and processing.

Respect to the reviewed literature, we propose a solution that goes from the RGB image, to the one with computed water level without any need for human intervention. Some of the reviewed works, proposed a solution able to deal with day frames only. We have developed a solution able to understand if the frame under analysis is a day or night one, and then apply a customized step of image processing. Respect to [14], where the solution is developed inside a laboratory, we developed and tested the solution on a very large real data set. Respect to [15] we took care of the night frames. Being flood monitoring a continuous task, we believe that solutions should take night-operation into consideration as we did. Generally, this system is going to be installed into unconstrained and not-standardised contexts, hence, it can't rely on strong chromatic assumptions, like the solutions proposed by [16], which requires the gauge to be brighter than its background, by [17], whose processing steps rely on color and morphology of the gauge, by [21] that exploits color for extracting gauge's ROI. Our solution is not based on chromatic assumptions. Other works proposed solutions based on teh manual definition of gauge's ROI. By analysing our data, we found out that oscillation of the camera can happen, mostly due to weather conditions like strong wind. Solutions based on an a-priori gauge's ROI (like those presented in [18–20]) are not reliable at all.

For what it concerns reliability during extreme weather, we specify that six events happened during the period of acquisition, resulting in tens of frames analysed and all water levels correctly computed, thus proving the goodness of the proposed solution. Moreover, one problem that can interfere with the correct functioning is the clearness of the water, that could make the underwater content visible. Even though it is very rare in the site analysed, it happened sometimes over the year of acquisition, and thanks to refraction and reflection it did not cause computing errors.

Furthermore, the proposed algorithm allows the adaptation in multiple sites with minimal effort, as proved by applying it on some samples related to thirteen additional sites, for which we have few images collected by volunteers. Once the V-IoT device is installed, a frame is snapped and rectification parameters for the site under analysis are set in few seconds. Then, some images are added to our ACF detector for adapting it to the additional site. After one or two days of data collection, site-specific thresholds for Module 1 and Module 2 are defined. Therefore, The adaptation required to the proposed solution is mostly regarding the initial system setup and, subsequently, the algorithm will be able to work on the new site autonomously and continuously. However, it is very important to set additional data collection campaigns, able to enlarge our data set to multiple sites, thus allowing us to test the reliability of the proposed solution in similar yet distinct contexts.

## 5. Conclusions

Based on results achieved and discussed, the proposed solution solved main limitations found in existing literature. It is therefore possible to move on to the next step, which is putting into operation in several sample sites. The work carried out proved to be very reliable, and an essential part is Module 1, the Image Category Classification. It is particularly necessary for V-IoT systems, especially for battery powered nodes, where data transmission is among the most consuming activities, and may be related to unusable data resulting in an energy waste. Ensuring the quality of data to be transmitted with a light and fast algorithm will help for having good data while extending system life.

Focusing on the second Module, results achieved are satisfactory, especially for what it concerns night frames as expected, since this category of frame is characterised by a strong standardisation of key characteristics. Day frames, connected to a very broad range of different light rays inclination respect to the camera objective, are more diversified and therefore critical to be analysed. For this reason, in future work we are going to test additional possibilities dedicated to improve the reliability on day frames. Respect to past works reviewed, we tested our algorithm on an extremely wide data set, which allowed us to be sure about its overall reliability respect to the task of measuring the water level.

**Author Contributions:** Conceptualization, L.S., L.P., A.B., F.S. and P.P.; methodology, L.S., L.P., A.B. and P.P.; software, L.S. and A.B.; validation, L.S., L.P., A.B., F.S. and P.P.; formal analysis, L.S., L.P., A.B., F.S. and P.P.; investigation, L.S., L.P., A.B., F.S. and P.P.; resources, L.S. and L.P.; data curation, L.S. and F.S.; writing—original draft preparation, L.S.; writing—review and editing, L.S., L.P. and A.B.; visualization, L.S.; supervision, L.P., A.B. and P.P.; project administration, P.P.; funding acquisition, P.P. All authors have read and agreed to the published version of the manuscript.

**Funding:** This research received no external funding.

**Data Availability Statement:** The data presented in this study are available on request from the corresponding author. The data are not publicly available due to the fact that the STREAM Project (see details hereafter) is under development.

**Acknowledgments:** This study has been promoted within the project entitled "STREAM—Strategic development of flood management", 2014–2020 Interreg V-A, Italy-Croatia CBC Programme-Call for proposal 2019, Priority Axis: Safety and resilience-Specific objective: Increase the safety of the Programme area from natural and man-made disaster, Application ID: 10249186.

**Conflicts of Interest:** The authors declare no conflict of interest.

**Sample Availability:** Samples of the images used for developing the work are available from the authors.

## Abbreviations

The following abbreviations are used in this manuscript:

| | |
|---|---|
| ML | Machine Learning |
| CV | Computer Vision |
| V-IoT | Visual-Internet of Things |

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
