# Peer review of "A Computer Vision System for Staff Gauge in River Flood Monitoring"

_inventions, doi:10.3390/inventions6040079_

Round 1

Reviewer 1 Report

The manuscript is well-written, but need to improve to some extent.

Please see my comments below-

Manuscript ID: inventions-1435350

Title: A Computer Vision System for Staff Gauge in River Flood Monitoring

Abstract:

Line 11: “V-Iot” could be abbreviated in line 7.

Line 11-13: “The results 12 show excellent performance in discerning bad quality frames from good quality both in day and 13 night scenario as well as in the water level detection.” – I would recommend to rephrase this sentence to make it clear.

Introduction

Line 72: “h24” or “24h”?

Line 90: “describe” or “described”?

Line 96: “They” – who? Not clear.

Line 110-113: Very big sentence, meaning not clear, I would recommend rephrasing the sentence.

Line 139 -144: I would prefer them combining them in a sentence rather than numbering.

Materials and methods

Well described, comprehensive.

Results

Sometimes you refereed “Fig”, and another time “figure”/ “Figure”. Should you be consistent?

Figure 10: Caption could be updated

Line 365 – 371: seems discussion and conclusion, but placed in results section.

Figure 12a and 12b: I saw “result achieved” stated 3 times in the captions. It could be reworded.

Discussion and conclusion

Discussion and conclusion could be separated instead of merged together. Particularly, the discussion section is deficit of information. I couldn’t see any comparison with relevant statements, agreement, disagreement, along with refs.

You could state your findings, compare with the advantages/ disadvantages of other methods with refs, and discuss the reason.

Specific usefulness of these findings along with the relevant literature (advantage/ disadvantage of this new finding) is required to clarify.

I would recommend updating this section.

Please improve the conclusion matching with your objectives.

Author Response

This is the list of reviewers' comments on the manuscript with the ID inventions-1435350, “A Computer Vision System for Staff Gauge in River Flood Monitoring”, submitted to Inventions. We thank the reviewers for their comments. This is the detailed authors' responses and explanations of changes done according to the reviewers' comments.  

The text is structured in the following manner.  

Each reviewer’s comment is numerated and written in a gray comment-heading field. Each reviewer's comment is numerated by two numbers – R.x.y, where x denotes the reviewer's ordinal number, and the y denotes the reviewers' comment ordinal number, for the reviewer x (if applicable).  

Response to the reviewer, and explicit action taken by the authors, are given below each reviewer's comment. Obviously, the references to the sections, pages, columns and lines included in the present document, are those related to the current version of the paper and not to the previous one. 

Finally, in the revised version of the paper, the actions took in response to the reviewer's comments have been added in blue.

We hope that you find our revisions adequate and our paper suitable for publication in Inventions 

Comment R.1.1: Line 11 “V-Iot” could be abbreviated in line 7. 

Response: We thank the reviewer for the suggestion.  

Action(s) taken: We have used the abbreviation at line 7. 

Comment R.1.2:  Line 11-13: “The results 12 show excellent performance in discerning bad quality frames from good quality both in day and 13 night scenario as well as in the water level detection.” – I would recommend to rephrase this sentence to make it clear. 

Response: We thank the reviewer for the opportunity to improve the quality of the paper. Action(s) taken: We have rephrased the sentence with the aim of making it clearer. 

Comment R.1.3: Line 72: “h24” or “24h”? 

Response: We thank the reviewer for the comment.  

Action(s) taken: We have updated the English form. 

Comment R.1.4: Line 90: “describe” or “described”? 

Response: We thank the reviewer for the comment. 

Action(s) taken: We aligned to the past form every reviewed paper. 

Comment R.1.5: Line 96: “They” – who? Not clear. 

Response: We thank the reviewer for the comment. 

Action(s) taken: We have changed the sentence in order to make the reference to the previous paper clearer. 

Comment R.1.6: Line 110-113: Very big sentence, meaning not clear, I would recommend rephrasing the sentence. 

Response: We thank the reviewer for the precious suggestion. 

Action(s) taken: We have changed the sentence in order to make the meaning clearer. 

Comment R.1.7: Line 139 -144: I would prefer them combining them in a sentence rather than numbering. 

Response: We thank the reviewer for the opportunity to improve the quality of the paper. We decided to proceed with the MDPI English editing service in order to refine the English of the paper.  

Action(s) taken: English language modifications of the entire paper according to the editing service have been inserted in the revised version of the paper. 

Comment R.1.8: Sometimes you refereed “Fig”, and another time “figure”/ “Figure”. Should you be consistent? 

Response: We thank the reviewer for the comment. 

Action(s) taken: We decided to align all the references to the figures using the “Fig.” form. 

Comment R.1.9: Figure 10: Caption could be updated 

Response: We thank the reviewer for the suggestion, even though it is not clear at all the final aim of caption update. 

Action(s) taken: We have updated the Caption trying to improve it. 

Comment R.1.10: Line 365 – 371: seems discussion and conclusion, but placed in results section. 

Response: We thank the reviewer for the comment. 

Action(s) taken: We have better positioned the sentence brought to our attention by the reviewer. 

Comment R.1.11: Figure 12a and 12b: I saw “result achieved” stated 3 times in the captions. It could be reworded. 

Response: We thank the reviewer for the comment. 

Action(s) taken: We removed repetitions from sub-captions. 

Comment R.1.12: Discussion and conclusion could be separated instead of merged together. Particularly, the discussion section is deficit of information. I couldn’t see any comparison with relevant statements, agreement, disagreement, along with refs. 

You could state your findings, compare with the advantages/ disadvantages of other methods with refs, and discuss the reason. 

Specific usefulness of these findings along with the relevant literature (advantage/ disadvantage of this new finding) is required to clarify. 

I would recommend updating this section. 

Please improve the conclusion matching with your objectives. 

Response: We thank the reviewer for the opportunity to improve the quality of the paper.  

Action(s) taken: We have divided discussion from conclusions. We have deepened the discussion according to the suggestions in the comment, and improved the two sections. 

Reviewer 2 Report

The authors have chosen an essential topic for the study of river floods and their monitoring. The key goal is to prevent threats to people's safety and damage in emergencies such as floods. Despite the significant expansion of the arsenal of new practical means of registration and control, developing effective measures and monitoring systems is prevalent.
As you know, water levels in the river have been recorded since the Pharaohs in ancient Egypt. Thus, nilometers were used to measure water transparency and water level in the Nile River during the annual flood season. Moreover, there were three main types of nilometers which were also calibrated in Egyptian cubits (a vertical column, a ladder leading to the water, a well with a water pipe)…
Over the past centuries, progress and created technologies have significantly improved the desired capabilities of monitoring river floods through automated means. In turn, the development of information technologies and Computer Vision, in particular, is proceeding at a very high pace.

The development of a highly reliable solution based on computer vision in this work boils down to two main modules. Undoubtedly, introducing one new element may sometimes be enough to confirm the novelty of a technical solution in a patent. However, when analyzing monitoring tasks as a highly reliable system, it is desirable to disclose its more complete description (implemented functions, modules of this new system). At the same time, the proposed review of the analogs of the object of research and the results of related research of the subject area in the materials of the article looks somewhat selective. It is proposed to illustrate the general control contour, for example, including all the intended consumers of the collected data (it is quite possible that unskilled personnel) in the considered extreme operating conditions of the solution being developed.

This will presumably allow us to indicate the "bottlenecks," the critical role and place for the evidentiary provision of not only the necessary but also sufficient reliability of elements (created from such elements of the system as a whole).

The cost factor will limit a comparative analysis of the best elements by the reliability criterion. The authors' choice of an acceptable compromise can be most advantageous here in an explicit form, for example, for threshold values of quantities. It is essential here that the general requirements for the system being developed in the article materials can supplement specific numerical quantities values.

These proposals are mainly advisory here, but the authors can improve the quality of the argumentation of the approach.

So, the goal of the work is to create a universal and intelligent automatic algorithm for measuring the water level, which will be further applied at several facilities with minimal adaptation efforts. The need to ensure scalability shortly is also highlighted. In this regard, it makes sense to briefly pay attention to what will achieve universality and scalability in the overall cost or the feasibility study, not in the form of assumptions but a series of arguments.

It is advisable to supplement the review with literature sources covering specific measurement algorithms (not only water level) in monitoring systems.

An important point is the possibility of a dedicated description of the original function or features of monitoring data processing. What is new, and what exactly is the authors' contribution (if any) in the areas under consideration (ML – Machine Learning, CV – Computer Vision, V-IoT – Visual-Internet of Things)?

At the authors' discretion, it is proposed here to pay attention to well-known applied related methods and models. In this case, they mainly relate to a specific area of hydrometry. Standardized procedures may be helpful (for example, among those often mentioned, including from the "Storm Water Management Model (SWMM), the United States Environmental Protection Agency (EPA)"). Moreover, the authors do not thoroughly analyze notable achievements both in the available articles of the MDPI publishing house and others (among others, it is proposed to independently review works similar to the previously mentioned – URL: https://www.mdpi.com/1424-8220/19/22/5012 ).

Thus, modern means of hydrological optimization with the use of new distributed monitoring systems have tremendous potential:

For extreme conditions, will this time interval of transmission of the collected data be acceptable for the emergency response specified in the article?

The authors claim that they tested the algorithm on an extensive data set. As far as possible, it makes sense to present them in any convenient form in an appendix to the materials of the submitted article.

In conclusion, – the statement about the sufficiency of the properties of the algorithm (procedure) in describing the qualities of a highly reliable monitoring system has a debatable character – how exactly will it be and will it undergo a change during the initial setup and scaling in a new distributed system?

These provisions do not reduce the overall favorable impression of the results of the research carried out by the authors on the development of basic modules and an algorithm for the functioning of the river flood monitoring system.

Author Response

This is the list of reviewers' comments on the manuscript with the ID inventions-1435350, “A Computer Vision System for Staff Gauge in River Flood Monitoring”, submitted to Inventions. We thank the reviewers for their comments. This is the detailed authors' responses and explanations of changes done according to the reviewers' comments.  

The text is structured in the following manner.  

Each reviewer’s comment is numerated and written in a gray comment-heading field. Each reviewer's comment is numerated by two numbers – R.x.y, where x denotes the reviewer's ordinal number, and the y denotes the reviewers' comment ordinal number, for the reviewer x (if applicable).  

Response to the reviewer, and explicit action taken by the authors, are given below each reviewer's comment. Obviously, the references to the sections, pages, columns and lines included in the present document, are those related to the current version of the paper and not to the previous one. 

Finally, in the revised version of the paper, the actions took in response to the reviewer's comments have been added in blue.

We hope that you find our revisions adequate and our paper suitable for publication in Inventions 

Comment R.2.1: The authors have chosen an essential topic for the study of river floods and their monitoring. The key goal is to prevent threats to people's safety and damage in emergencies such as floods. Despite the significant expansion of the arsenal of new practical means of registration and control, developing effective measures and monitoring systems is prevalent. As you know, water levels in the river have been recorded since the Pharaohs in ancient Egypt. Thus, nilometers were used to measure water transparency and water level in the Nile River during the annual flood season. Moreover, there were three main types of nilometers which were also calibrated in Egyptian cubits (a vertical column, a ladder leading to the water, a well with a water pipe)…Over the past centuries, progress and created technologies have significantly improved the desired capabilities of monitoring river floods through automated means. In turn, the development of information technologies and Computer Vision, in particular, is proceeding at a very high pace. The development of a highly reliable solution based on computer vision in this work boils down to two main modules. Undoubtedly, introducing one new element may sometimes be enough to confirm the novelty of a technical solution in a patent. However, when analyzing monitoring tasks as a highly reliable system, it is desirable to disclose its more complete description (implemented functions, modules of this new system). At the same time, the proposed review of the analogs of the object of research and the results of related research of the subject area in the materials of the article looks somewhat selective. It is proposed to illustrate the general control contour, for example, including all the intended consumers of the collected data (it is quite possible that unskilled personnel) in the considered extreme operating conditions of the solution being developed. This will presumably allow us to indicate the "bottlenecks," the critical role and place for the evidentiary provision of not only the necessary but also sufficient reliability of elements (created from such elements of the system as a whole). The cost factor will limit a comparative analysis of the best elements by the reliability criterion. The authors' choice of an acceptable compromise can be most advantageous here in an explicit form, for example, for threshold values of quantities. It is essential here that the general requirements for the system being developed in the article materials can supplement specific numerical quantities values.  

These proposals are mainly advisory here, but the authors can improve the quality of the argumentation of the approach. 

Response: We thank the reviewer for the detailed comments. We would like to express our agreement with the proposed “food-for-taught". 

Action(s) taken: Without interfering too much with the aim of the paper, which is to propose a versatile CV solution for V-IoT systems dedicated to water level monitoring, we tried to highlight arguments in line with the proposed reflections, whenever possible. 

Comment R.2.2: So, the goal of the work is to create a universal and intelligent automatic algorithm for measuring the water level, which will be further applied at several facilities with minimal adaptation efforts. The need to ensure scalability shortly is also highlighted. In this regard, it makes sense to briefly pay attention to what will achieve universality and scalability in the overall cost or the feasibility study, not in the form of assumptions but a series of arguments. 

Response: We thank the reviewer for the opportunity to clarify the field of action of the proposed work. We can’t provide detailed financial information, since this work arise from a big research project, which involves several collaborating partners, and each partner has specific duties. The Authors and the Università Politecnica delle Marche are technological partner, in charge of research and implementation. Feasibility in terms of money is involved in the project but it is not our responsibility. The partner in charge, has defined hardware budget based on technical specifications set by us based on this preliminary work which has ensured the overall technical feasibility of the solution. 

Action(s) taken: We have answered to the reviewer comment. 

Comment R.2.3: It is advisable to supplement the review with literature sources covering specific measurement algorithms (not only water level) in monitoring systems. 

Response: We thank the reviewer for the suggestion. We have carefully evaluated the possibility to enlarge the scope of the reviewed literature, adding measurement algorithms in monitoring systems (also of other types with respect to hydrometric level monitoring), but we concluded that such an extension could mislead readers in comparison of paper’s focus and goal. We specify that, as previously mentioned, this work is part of the broader STREAM research project funded by Interreg Italia Croatia. We envision the integration of this piece of research into a wider Early Warning System (EWS) for Flood hazards, involving both water level measurement, satellite images analysis, and other information to be fused in order to implement a reliable EWS.  

Action(s) taken: We answered to the reviewer. 

Comment R.2.4: An important point is the possibility of a dedicated description of the original function or features of monitoring data processing. What is new, and what exactly is the authors' contribution (if any) in the areas under consideration (ML – Machine Learning, CV – Computer Vision, V-IoT – Visual-Internet of Things)? 

Response: We thank the reviewer for the opportunity to improve the quality of the paper.  

Action(s) taken: We have divided discussion from conclusions and inserted more details on each of the two sections, highlighting novelty of the proposed solution.  

Comment R.2.5: At the authors' discretion, it is proposed here to pay attention to well-known applied related methods and models. In this case, they mainly relate to a specific area of hydrometry. Standardized procedures may be helpful (for example, among those often mentioned, including from the "Storm Water Management Model (SWMM), the United States Environmental Protection Agency (EPA)"). Moreover, the authors do not thoroughly analyze notable achievements both in the available articles of the MDPI publishing house and others (among others, it is proposed to independently review works similar to the previously mentioned – URL: https://www.mdpi.com/1424-8220/19/22/5012 ). 

Response: We thank the reviewer for the opportunity to improve the quality of the paper and for the suggested procedures like SWMM. We decided not to enlarge the scope of our paper, being focused on proposing a solution for water level computation through the analysis of frames captured by cameras installed in river sites. 

Action(s) taken: We have included two additional relevant references (regarding Computer Vision-based water level computation solutions) in our paper, inside the Introduction. 

Comment R.2.6: Thus, modern means of hydrological optimization with the use of new distributed monitoring systems have tremendous potential: For extreme conditions, will this time interval of transmission of the collected data be acceptable for the emergency response specified in the article? 

Response: We thank the reviewer for the comment. We specify that the transmission interval related to the data set used for developing the solution is 30 minutes, but once put in service we fixed the time interval between frames at 3 minutes. This obviously requires more transmission band, but thanks to Module 1 the overall optimization of data transmission is ensured (only if the frame snapped is an analyzable frame it is sent to the central cloud server for processing based on Module 2) 

Action(s) taken: We better outlined inside Discussion Section the difference between the transmission interval used for developing the large data set, and the envisioned transmission interval once the solution will be put in place. 

Comment R.2.7: The authors claim that they tested the algorithm on an extensive data set. As far as possible, it makes sense to present them in any convenient form in an appendix to the materials of the submitted article. 

Response: We thank the reviewer for the comment. This work is developed within the broad STREAM project funded by Interreg Italy-Croatia. The partner owning the data used for developing this solution made them available to us in the face of NDA signing. Data are therefore available to people asking to us and willing to sign the same NDA we signed. We specify that, once completed the project and put the solution in place, the Civil Protection will make all the data available through their website.  

Action(s) taken: We answered to the reviewer. 

Comment R.2.8: In conclusion, – the statement about the sufficiency of the properties of the algorithm (procedure) in describing the qualities of a highly reliable monitoring system has a debatable character – how exactly will it be and will it undergo a change during the initial setup and scaling in a new distributed system? 

These provisions do not reduce the overall favorable impression of the results of the research carried out by the authors on the development of basic modules and an algorithm for the functioning of the river flood monitoring system. 

Response: We thank the reviewer for the opportunity to improve the quality of the paper.  

Action(s) taken: We have better described inside the Discussion section the initial setup envisioned for the put in place of the solution. 

Round 2

Reviewer 1 Report

Dear authors

Thank you for updating the manuscript. it is now more comprehensive, well written.

Reviewer 2 Report

Thanks to the authors for the response provided to suggestions for improving the materials of the article.

All answers in reasonable arguments are justified on points, and the authors' considered opinion is taken into account.

Considering the expected final stage of the design of the materials of the article, I believe that in this form, the totality of the main results will look balanced to the extent necessary for publication.

With best wishes for success in the upcoming implementation of this and subsequent elements of systems for monitoring, controlling, and eliminating consequences of emergencies, including the considered technological solutions of computer vision for monitoring river floods.